# Network Analysis of Depression Using Magnetoencephalogram Based on Polynomial Kernel Granger Causality

**DOI:** 10.3390/e25091330

**Published:** 2023-09-13

**Authors:** Yijia Ma, Jing Qian, Qizhang Gu, Wanyi Yi, Wei Yan, Jianxuan Yuan, Jun Wang

**Affiliations:** 1Smart Health Big Data Analysis and Location Services Engineering Research Center of Jiangsu Province, School of Geographic and Biologic Information, Nanjing University of Posts and Telecommunications, Nanjing 210023, China; 2School of Communications and Information Engineering, Nanjing University of Posts and Telecommunications, Nanjing 210023, China; 3Department of Psychiatry, Affiliated Nanjing Brain Hospital, Nanjing Medical University, Nanjing 210029, China

**Keywords:** brain network, magnetoencephalogram, Granger causality, kernel function

## Abstract

Depression is a psychiatric disorder characterized by anxiety, pessimism, and suicidal tendencies, which has serious impact on human’s life. In this paper, we use Granger causality index based on polynomial kernel as network node connectivity coefficient to construct brain networks from the magnetoencephalogram (MEG) of 5 depressed patients and 11 healthy individuals under positive, neutral, and negative emotional stimuli, respectively. We found that depressed patients had more information exchange between the frontal and occipital regions compared to healthy individuals and less causal connections in the parietal and central regions. We further analyzed the topological properties of the network revealed and found that depressed patients had higher average degrees under negative stimuli (*p* = 0.008) and lower average clustering coefficients than healthy individuals (*p* = 0.034). When comparing the average degree and average clustering coefficient of the same sample under different emotional stimuli, we found that depressed patients had a higher average degree and average clustering coefficient under negative stimuli than neutral and positive stimuli. We also found that the characteristic path lengths of patients under negative and neutral stimuli significantly deviated from small-world network. Our results suggest that the analysis of polynomial kernel Granger causality brain networks can effectively characterize the pathology of depression.

## 1. Introduction

Depression is an emotional disorder mainly characterized by persistent sadness and a loss of interest. It’s always accompanied by changes in cognitive function and biology, failure to regulate negative emotions is one of the main factors of depression [1,2,3]. Today, depression has become one of the most common neuropsychiatric disorders characterized by anxiety, pessimism, and suicidal thoughts which seriously affects human’s life and poses challenges to social development [4,5].

Human brain is an extremely complex system, whose precise structure and efficient functional connection mode provide the basis for the differentiation and integration of brain information [6]. Complex networks based on graph theory can effectively describe and evaluate the function and structure of the brain and are widely used in the study of the pathogenesis of mental illness. Researchers found that the construction of a functional brain network and the analysis of topological properties using the data obtained by electroencephalogram (EEG), magnetoencephalogram(MEG), and Functional magnetic resonance imaging (fMRI) can provide brain network imaging markers for the diagnosis and treatment of mental illness [7,8,9]. Morabito et al. [10] studied the resting-state EEG of patients with Alzheimer’s disease (AD) and constructed a brain network. They calculated brain network parameters, such as feature path length, clustering coefficient, and small-world attributes, and found that brain efficiency decreased and small-world connectivity was lost. Depression is a common mental disease which is closely linked to abnormalities in brain function and structure [11]. Previous studies have shown that there are abnormalities in the global topological properties and regional connections of complex brain networks in depressed patients. The complex brain network may provide us with quantitative indicators for the diagnosis and treatment of depression and provide an objective basis for the physiological and pathological mechanism of depression.

The Granger causality is a common tool used for analyzing linear time series. After the Granger causality was proposed in 1969, it has been widely used in economics, physiology, neuroscience, and other fields [12,13,14,15]. The actual physiological signals are always nonlinear in neuroscience [16]. The traditional Granger causality cannot accurately analyze the internal relationships between nonlinear signals. Mapping low-dimensional nonlinear data to high-dimensional spaces to make them linearly separable can effectively solve this problem, causing a “dimension disaster” in high-dimensional space computing. The kernel function computes a similarity measure in feature space by vector dot product, which can solve complex calculation problems in a high-dimensional space [17]. Therefore, combining a kernel function with linear Granger causality can effectively analyze the nonlinear time series.

MEG is a kind of physiological signal with non-linear characteristics, the different MEG channels can be used as nodes to construct a brain network to observe the information exchange between different brain regions. Behrad [18] proposed a network-localized Granger causality (NLGC) paradigm to directly estimate its parameters from MEG measurements. They use NLGC to process MEG data of young and elderly participants with auditory tasks. Changes in the interactions between different regions of the brain in the resting and tone-processing states of the brain were discovered in the δ bands. In our study, we use the Granger causality index based on the polynomial kernel as a measure of the connection edge between the different brain nodes to construct brain networks for depressed patients and healthy people. Then further analyze the topological properties of network, which aims to help with the clinical diagnosis and treatment of depression.

## 2. Methods

### 2.1. Granger Causality Basic Theory

In order to complete the transformation from linear Granger causality to non-linearity, it is necessary to embed the data into the Hilbert space for operation [19]. It is specifically expressed as follows:

Suppose that there are two random variables, *X*, *Y*; let $X_{i}={\left(\xi_{i},\ldots ,\xi_{i+m-1}\right)}^{T}$, $Y_{i}={\left(\eta_{i},\ldots ,\eta_{i+m-1}\right)}^{T}$, $x_{i}=\xi_{i+m}$, $X_{i}$,$Y_{i}$,$x_{i}$ is the realization of *X*, *Y* and *x*, respectively. $X_{i}$ is the column vector of *X*, and *X* is a m∗N matrix. Define $Z_{i}=\left(X_{i}^{T},Y_{i}^{T}\right)$, so $Z_{i}$ is a 2m∗N matrix. Suppose $x^{T}x=1$, and that the mean of each component of the random variables *X* and *Y* is 0, then for i=1,2,...,N, we have the following:(1)$${\tilde{x}}_{i}=\sum_{j=1}^{m}A_{j}\xi_{i+m-j}$$
(2)$$x_{i}=\sum_{j=1}^{m}A_{j}\xi_{i+m-j}+\sum_{j=1}^{m}B_{j}\eta_{i+m-j}$$
The values of vectors $\tilde{x}={\left({\tilde{x}}_{1},\ldots ,\tilde{x_{N}}\right)}^{T}$ can be estimated by the linear regression method. $K=X^{T}X$ is an N∗N matrix, let $H\subseteq {ℜ}^{N}$ be in the range of *K*. Let the eigenvector of *K* be $v_{1},...,v_{n}$ and satisfy the orthogonal relation. For this eigenvector, there are
(3)$$P=\sum_{i=1}^{m}v_{i}v_{i}^{T}$$$x^{\prime }$ is the projection of *x* on *H*, we have $\tilde{x}=Px$,y=x−Px. Similarly, $H^{\prime }$ is in the range of $K^{\prime }=Z^{T}Z$ and $x^{\prime }=P^{\prime }x$. Here, $H\subseteq H^{\prime }$, we decompose $H^{\prime }$ into $H⊕H^{⊥}$, $H^{⊥}$ denotes all spaces in $H^{\prime }$ that are orthogonal vectors in space *H*. The projection of *P* in space $H^{⊥}$ is expressed as $P^{⊥}$, and the Granger causality intensity is obtained:(4)$$\delta =\frac{{∥P-y∥}^{2}}{1-{\tilde{x}}^{T}\tilde{x}}$$All nonzero eigenvectors of the matrix $\tilde{K}$ are extended, the eigenvectors are expressed as $t_{1},...,t_{m}$, and $H^{⊥}$ is the set of the eigenvectors. The range of values for $\tilde{K}=K^{\prime }-PK^{\prime }-K^{\prime }P-PK^{\prime }P$, $H^{⊥}$ is in the range of $\tilde{K}$. R is the Pearson correlation coefficient of *y* and $t_{i}$ is $r_{i}$; we have:(5)$$\delta =\sum_{i=1}^{m}r_{i}^{2}$$Finally, the Granger causality index with the filtering property is obtained by the FDR correction:(6)$$\delta_{F}=\sum_{i}r_{i}^{2}$$

### 2.2. Granger Causality Based on Polynomial Kernel

The kernel function of a non-homogeneous polynomial of order *p* (*p* is an integer) is defined as
(7)$$K_{p}\left(X,X^{\prime }\right)={\left(1+X^{T}X^{\prime }\right)}^{p}$$In this case, the characteristic function is unary, and the input variable is up to the order of *p*. The dimension of the *H* space is $m_{1}=1/B\left(p+1,m+1\right)-1$; when *p* = 1, it is a linear regression. The dimension of the $H^{\prime }$ space is $m_{2}=1/B\left(p+1,2m+1\right)-1$. Here, $H\subseteq H^{\prime }$, $H^{\prime }$ into $H⊕H^{⊥}$. The calculation formula is $\tilde{K}=K^{\prime }-PK^{\prime }-K^{\prime }P-PK^{\prime }P$. We construct the kernel Granger causality corrected by the FDR to obtain the following:(8)$$\delta_{F}^{K}=\sum_{i^{\prime }}r_{i^{\prime }}^{2}$$

### 2.3. Introduction to Topology Properties of Brain Network

Degree:The degree $D_{i}$ of a node is defined as the number of edges connected to node *i* in the network, and the degree of the i node can be expressed as follows:
(9)$$D_{i}=\sum_{i}C_{ij}$$
where $C_{ij}$ is the state of connection between nodes *i* and *j*. When there is a connection between *i* and *j*, $C_{ij}$ = 1. When there is no connection between *i* and *j*, $C_{ij}$ = 0. When the value of *i* increases, the more connected edges of the node, representing the node, became more important. The average degree of the network is determined by the mean of the degree of all nodes. Average degree reflects the connection level between nodes and measures the complexity of the network.Clustering coefficient:The clustering coefficient *C* represents the clustering situation of the nodes in the network, which means *C* represents the probability that the node neighbors are also neighbors to each other. The clustering coefficient $C_{i}$ of node *i* can be defined as follows:
(10)$$C_{i}=\frac{2e^{i}}{k_{i}\left(k_{i}-1\right)}$$Here, $e_{i}$ represents the number of connections between $k_{i}$ nodes connected to node *i*, $k_{i}\left(k_{i}-1\right)$ represents the maximum number of connected edges between $k_{i}$ nodes connected to node i. The average clustering coefficient of a network is the mean of the clustering coefficient of all nodes in the network, which can be expressed as follows:
(11)$$C=\frac{\sum_{i}^{N}C_{i}}{N}$$*N* is the number of nodes in the network and the value of $C_{i}$ is between 0 and 1. When a connection exists between both nodes in the network, C=1, and when all nodes in the network do not have a connection, C=0. The clustering coefficient can reflect the tightness of connections between nodes.Average path length:The calculation formula for the average path length *L* of the network is
(12)$$L=\frac{1}{N(N-1)}\sum_{i\neq j\in N}d_{ij}$$
where N is the number of nodes in the network, $d_{ij}$ is the shortest distance between node *i* and node *j*, representing the minimum number of connection edges required for connectivity between node i and node j. In small-scale networks, we usually use the Floyd algorithm to calculate $d_{ij}$. The average path length is an important indicator to measure the transmission efficiency of network. Random networks typically have shorter average lengths, while regular networks typically have longer average path lengths.

## 3. Experimental Results and Analysis

### 3.1. Experimental Data

We collected the MEG data from the magnetoencephalography center of the Brain Hospital of Nanjing Medical University. The equipment used for collecting MEG was the Canadian CTF 275 full-head magnetoencephalogram acquisition system, and the sampling frequency was 1200 HZ. We have 16 samples including 11 healthy individuals and 5 depression patients. All participants had no bad habits and their physiological indicators were normal. The ages of participants were between 20 and 30 years, and the average age was 25 ± 2. All patients were evaluated by professional psychiatrists using the Hamilton Depression Rating Scale, consisting of 17 items (HDRS-17), and based on the International Classification of Diseases, 10th Revision (ICD-10). The patients were taking antidepressants during the test and had no other mental or neurological disorders. Their HDRS-17 scores were greater than 17. Before the experiment, we introduced the equipment to participants to ensure that they had no fear of the device. We obtain 240 images from the International Emotional Image Library (IAPS), consisting of 80 positive, 80 neutral, and 80 negative emotional images. Three types of images were displayed to participants consecutively in order during the experiment. Then we use the CTF275 MEG system to record the brain MEG signals of the 16 samples under different stimuli.

### 3.2. Experimental Process and Steps

We use MATLAB2022b to process the MEG data. The MEG data were pre-processed by artifact removal, baseline correction, and SPM8 filtering. The specific processing was as follows: for each data, a bandpass filter was used to obtain a signal from 0.1 HZ–200 HZ, then remove 50 HZ of industrial frequency noise, and we finally obtain a three-dimensional matrix of 275 × 161 × 80. 275 represents the 275 MEG channels, 161 represent the data sampling points, and 80 represents the number of each type of picture. The duration of the stimulus was 600 ms, with intervals of 650 ms to 800 ms, and the three types of pictures were randomly displayed on the screen in front of the subjects. In this experiment, we first analyzed the subjects’ δ-wave (0.5–4 HZ), θ-wave (4–8 HZ), α-wave (8–13 HZ), β-wave (13–30 HZ), and full frequency band using polynomial kernel Granger causality. We found significant differences in the β-band brain MEG signals (13–30 HZ) of the two groups. In Basar [20,21] et al.’s study on the response amplitude of different bands in the human EEG under facial expression stimuli, they found that the β-band response amplitude was significantly increased when subjects processed facial emotion picture stimuli. Compared to other bands, β-band was most active and negative emotional stimuli will trigger higher beta oscillations. Based on this, the following studies were conducted with beta-band EEG signals.

The brain can be divided into five brain regions, namely the frontal region (F), parietal region (P), occipital region (O), temporal region (T), and central region (C). As shown in Figure 1, the CTF 275 magnetoencephalography system further divides these five brain regions into left and right, or left, middle, and right.

When selecting the embedding dimension (m) of the polynomial kernel function, since the Granger causality is based on the theory of the regression model, we use the AIC (Akaike information criteria) when selecting the embedding dimension *m*. AIC is a method proposed by Akaike [22] et al. to obtain the order of a time series analysis model. While AIC encouraging good data fitting, it is necessary to avoid overfitting as much as possible, so that the model with the minimum AIC function is considered the optimal model. The expression of the AIC criterion used in this experiment is [23]:(13)$$AIC\left(m\right)=\ln\left(\sum \right)+\frac{2p^{2}m}{n}$$
where *p* is the dimension of the variable, ∑ is the covariance matrix of the prediction error obtained by *m* order regression, and *n* is the length of the time series. In this experiment, we determined order m of the model using the AIC information criterion and calculated the order of the model from 1 to 20. We found that when *m* = 4, both groups of people had smaller AIC values. Therefore, we chose to embedding dimension m=4. When determining the value of *p*, two groups of channels were randomly selected in the occipital and frontal regions of depressed patients and healthy people, and *p* = 1–6 was taken for calculation. As we can see in Figure 2, the polynomial kernel Granger causality strength between the two groups of channels peaked when *p* = 2. To avoid overfitting issues resulting from excessive index selection, we selected *p* = 2 for subsequent analyses.

Fixing m = 2 and p = 4, we used polynomial kernel Granger causality to calculate the connection strength between the two groups of 14 nodes two-by-two under three different emotional stimuli, and then averaged the two groups separately. Finally, we obtained the average Granger causality index matrix for the two populations under the three emotional stimuli and performed independent samples *t*-tests on them. The results are shown in Table 1. From the table, we can see that the mean of the Granger causality matrix is greater in depressed than in healthy individuals and there is a significant difference between depressed patients and healthy people (*p* < 0.05) under negative and neutral stimuli. However, there is no significant difference between the two groups of people under positive stimuli.

Before constructing a brain network, we need to select a suitable threshold value. The selection of the threshold (T) has been a difficult point in brain network research. It can directly affect the statistics and analysis of topological properties in brain networks. There is no universally applicable threshold selection method, and the following three principles will be referred to for threshold selection in this paper [24,25]: (1) The threshold value shall be chosen to ensure the complete connectivity of the network as much as possible, which means that there are no or very few isolated nodes in the network. (2) The threshold must be chosen to ensure the small-world nature of the network as much as possible to make the network has high global efficiency and low local efficiency. (3) The thresholds are selected to reflect the differences between groups as much as possible. Based on the above selection principle, we determined the threshold range of 0.11–0.13, and took a threshold every 0.001 for calculation. Finally we select T = 0.115. To further observe the specific differential areas of the brain of the two populations under negative and neutral stimuli, we draw the adjacency matrix of the difference between healthy and depressed individuals. As shown in Figure 3, blue squares indicate the presence of causal effects between two brain regions corresponding to the given coordinates in depressed individuals, with no connections observed in healthy individuals. Yellow squares signify the existence of causal effects between two brain regions corresponding to the coordinates for healthy individuals, with no connections found in depressed individuals. Green squares denote consistency between the two populations for the brain regions represented by the coordinates, i.e., either both groups exhibit a causal effect, or neither does.

As can be seen Figure 3, depressed patients had more causal interactions with other regions in RF (right frontal), LO (left occipital) and RO (right occipital) than healthy individuals under negative stimulation. Healthy individuals had more causal interactions with other regions in the LP (left parietal) and LC (left central) regions. Under neutral stimulation, depressed patients had more causal interactions in the LO (left occipital), RF (right frontal), RO (right occipital). In contrast, healthy people showed more causal interactions in the LP (left parietal) and ZC (central region) areas. These results suggest that under non-positive emotional stimuli, depressed individuals show more causal connections and closer information exchanges with other brain regions in the frontal and occipital regions compared to healthy individuals, and less causal connections in the parietal and central regions. The frontal lobe is closely related to the regulation of human emotions, and depressed patients have significant abnormalities in the frontal lobe compared to healthy individuals when processing negative stimuli [26]. The occipital lobe, as the primary processing area of the brain for visual stimuli, is mainly involved in the processing of facial emotions, and it has been shown that depressed patients have abnormal activation of the occipital lobe in response to negative facial emotional stimuli [27]. Furthermore, excessive attention to negative stimuli exists in the occipital region of depressed patients. The parietal region is mainly responsible for the integration of visual and spatial information, and some studies have shown that healthy people have higher neurotransmitter activation in the parietal region than depressed patients. Neurotransmitters are the main substances that transmit information between neurons, and a high activation of neurotransmitters in the parietal regions of healthy people indicates that healthy people have more information exchange and more intense activity in the parietal region, connecting with other brain areas [28]; This finding is consistent with our results, which conclude that healthy people have more causal connection in the parietal regions.

Based on the above data, we constructed brain networks with T = 0.115 for the MEG signals of depressed patients and healthy individuals, respectively, as shown in Figure 4, Figure 5 and Figure 6. From the figure, we can see that the connections of the brain network are tighter in depressed patients compared to healthy individuals under negative and neutral stimuli. Due to the large number of nodes, we cannot visually observe the topological properties of the network from the network diagram, so we will find changes in the topological properties of the two groups under different emotional stimuli by analyzing the average degree of the network, the average clustering coefficient, and the characteristic path length index in the small-world property.

Figure 7 and Figure 8 show the error bar graphs of the average degree and average clustering coefficients of the two populations under the three emotional stimuli: negative, neutral, and positive, respectively. In Figure 7, we can see that the average degree of depressed patients was higher than healthy individuals under negative and neutral stimuli, and both passed the independent samples *t*-test (*p* = 0.008). However, the two groups didn’t show significant differences under positive stimuli. This result suggests that the connections between brain network nodes are stronger in depressed patients under non-positive emotional picture stimuli and the causal connection between brain regions are significantly enhanced in depressed patients when processing negative stimuli. From Figure 8, we can see that the average clustering coefficients are higher in healthy individuals under all three emotional stimuli, and all pass the independent samples *t*-test with *p* = 0.034 under negative stimuli, *p* = 0.028 under neutral stimuli, and *p* = 0.011 under positive stimuli, indicating healthy individuals have a higher degree of brain network grouping. Next, we compare differences in mean degree and mean clustering coefficients for the same sample across emotional stimuli. From Figure 9, it can be seen that depressed patients show the highest average degree under negative stimuli and the lowest under positive stimuli. In contrast, healthy individuals show no significant differences under the three emotional stimuli. To verify whether there is a statistically significant difference in the average degree of the same sample under three different emotional stimuli. We conducted a one-way ANOVA analysis and set the significance level as α = 0.05. The original hypothesis H0 is: there is no significant difference in the average degree of the same type of sample under three emotional stimuli. After the calculation, it was found that the significant difference (*p*-value) in the average degree between healthy individuals under three different emotional stimuli was greater than 0.05, so there was no significant difference between the average degrees of the three emotional stimuli in healthy individuals. However, in depressed patients, the *p*-value was less than 0.05. This indicates that the differences in the average degree between the three stimuli are significant. We further used the Bonferroni multiple comparison test to correct results, we found that there were significant differences between positive and neutral stimuli (*p* = 0.025), positive and negative stimuli (*p* = 0.011), and neutral and negative stimuli (*p* = 0.041) in depressed patients. As can be seen in Figure 10, depressed patients had the largest average clustering coefficient under negative stimuli and the smallest under positive stimuli, while the average clustering coefficient did not change in healthy individuals under different emotional stimuli. We used the same method above to test whether there were significant differences in the average clustering coefficients of the same sample types. Finally, we found that there was no significant difference in the average clustering coefficients between healthy individuals under three emotional stimuli. However, there is significant differences in depressed patients between positive and negative stimuli (*p* = 0.024). This result suggests that the brain networks of depressed patients differ more under different emotional stimuli, which may be due to the fact that depressed patients are more prone to mood swings compared to healthy individuals.

A small-world network is a network with a high clustering coefficient and short characteristic path length, and the small-world property can be characterized by calculating the small-world coefficient. The characteristic path length of a network is the average of the shortest path lengths that all nodes in the network need to travel to connect. In this paper, we observed the changes in brain networks of depressed and healthy people by calculating the deviation in the characteristic path length of the network under positive, neutral, and negative stimuli compared to the characteristic path length of a random small-world network model. We randomly construct fifty small-world networks and calculate their characteristic path lengths (Ls). The mean path lengths (L) were calculated for depressed patients and healthy people under three emotional stimuli at different thresholds, and the degree of deviation of the two populations under different emotional stimuli compared to small-world networks was observed by calculating the value of L/Ls. If the value of L/Ls is closer to 1, it indicates that the characteristic path length of the network is close to that of the small-world network.

Figure 11 shows the comparison of the L/Ls values of the two groups under three different emotional stimuli, * represents *p* < 0.05 after using the independent samples *t*-test. It can be seen from the figure that the L/Ls values of the brain network of depressed patients are greater than 1 under both negative and neutral stimuli at different thresholds and have a tendency to continue to increase, indicating that the characteristic path length of the brain network of depressed patient has a significant deviation from a small-world property. On the contrary, the L/Ls values of the brain networks of depressed patients and healthy individuals did not differ significantly under positive stimuli and were close to 1, indicating that the brain networks of the two groups were close to the small-world network under positive stimuli. These results indicate that the characteristic path lengths of the brain networks of depressed patients significantly deviated from a small-world property under negative and neutral stimuli, and changes in the brain network of depressed patients were more pronounced, while the brain networks of healthy individuals were close to the small-world network properties under the three emotional stimuli.

## 4. Summary

In this paper, we used the Granger causality based on the polynomial kernel as the connecting edge between the brain network nodes to construct brain networks for two groups of people (depressed patients and healthy people); we divided their topological properties and obtained the following conclusions: (1) Under negative and neutral emotional stimuli, depressed patients exchanged information more closely between the frontal and occipital regions compared to healthy people, and the causal connection was weaker in the parietal and central regions. (2) Depressed patients had higher average degrees under negative and neutral emotional stimuli than healthy individuals, and healthy individuals had higher average clustering coefficients under all three emotional stimuli than depressed patients. (3) Comparing the changes of the mean degree and mean clustering coefficients in the same population under different stimuli, it was found that the mean degree and mean clustering coefficients were greater in depressed patients under negative stimuli than in neutral and positive stimuli, indicating that the brain networks of depressed patients changed more when they were subjected to emotional stimuli. (4) The mean path lengths of depressed patients deviated significantly from the small-world properties under negative and neutral stimuli. In conclusion, using Granger causality based on the polynomial kernel to analyze MEG signals and construct networks can effectively differentiate between healthy individuals and depressed patients, aiding in the diagnosis of depression. Finally, we would like to emphasize that larger and more representative patients are necessary for the validation of our findings; we will further increase our sample size and conduct more research on the topological structure and attributes of the brain network for depression and other mental disorders.

## Figures and Tables

**Figure 1 entropy-25-01330-f001:**
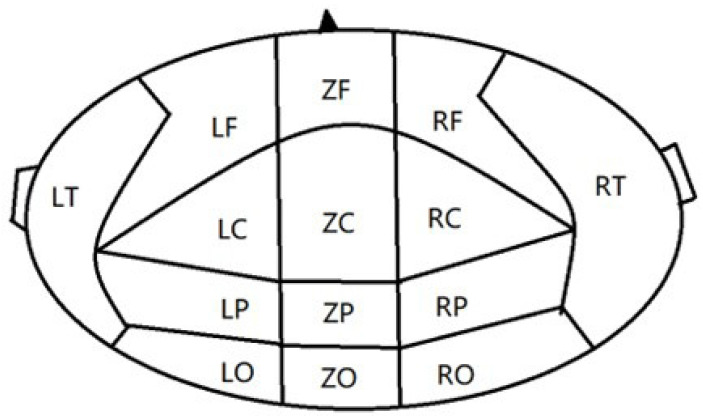
Dividing regions in the CTF 275 and the number of channels in each brain region.

**Figure 2 entropy-25-01330-f002:**
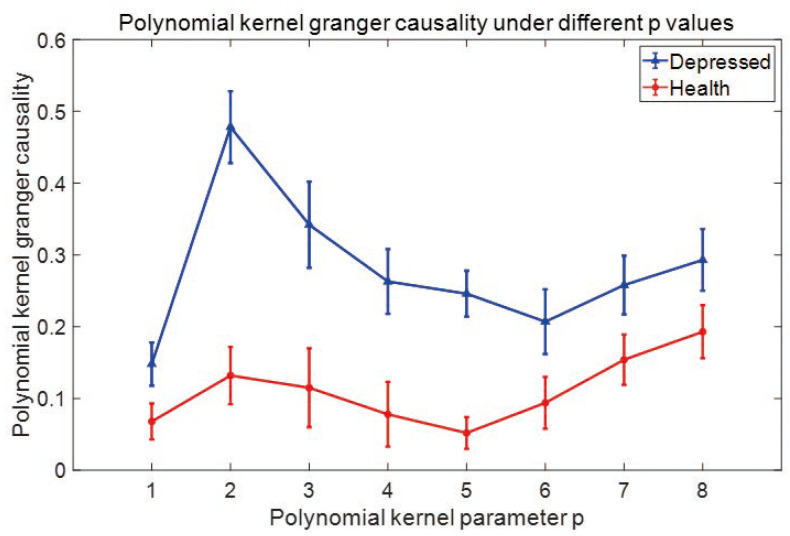
Selection of parameter *p* value.

**Figure 3 entropy-25-01330-f003:**
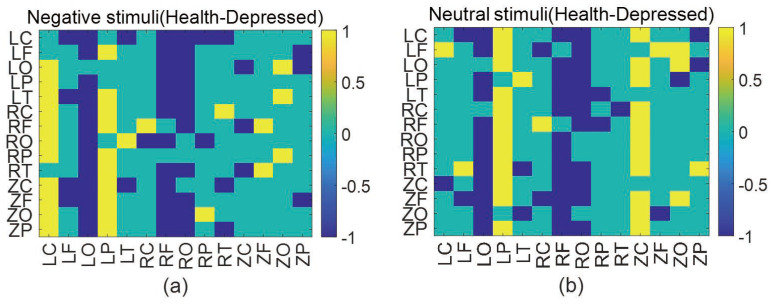
The diff-adjacency matrix between the two groups under negative and neutral stimuli. (**a**) negative stimuli (**b**) neutral stimuli.

**Figure 4 entropy-25-01330-f004:**
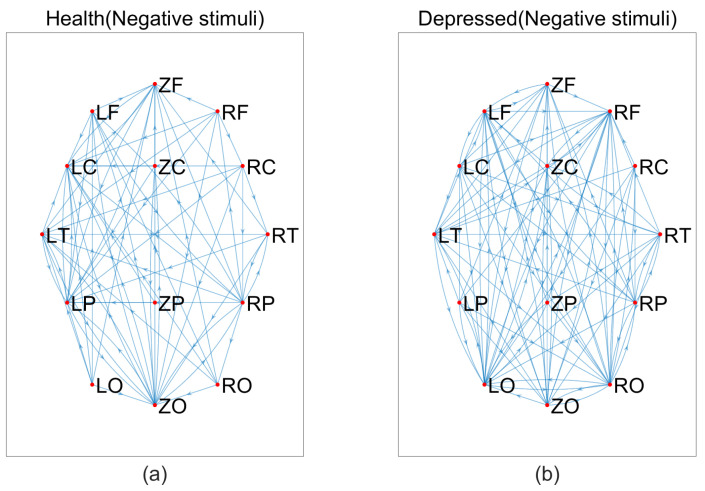
Brain networks in two groups under negative stimuli, (**a**) healthy, (**b**) depressed.

**Figure 5 entropy-25-01330-f005:**
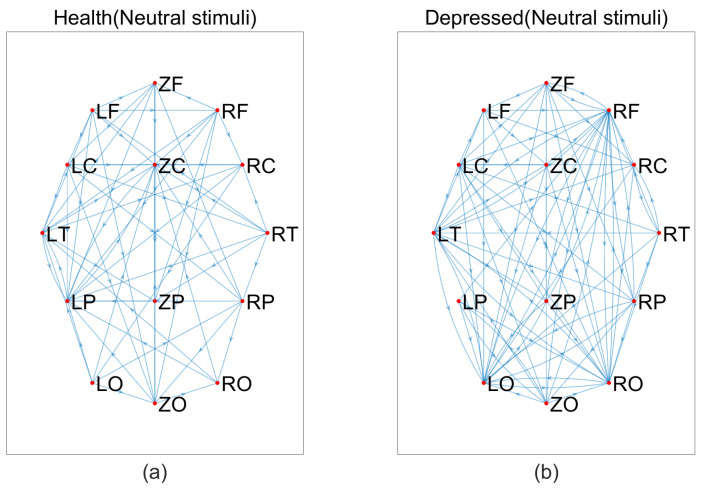
Brain networks in two groups under neutral stimuli, (**a**) healthy, (**b**) depressed.

**Figure 6 entropy-25-01330-f006:**
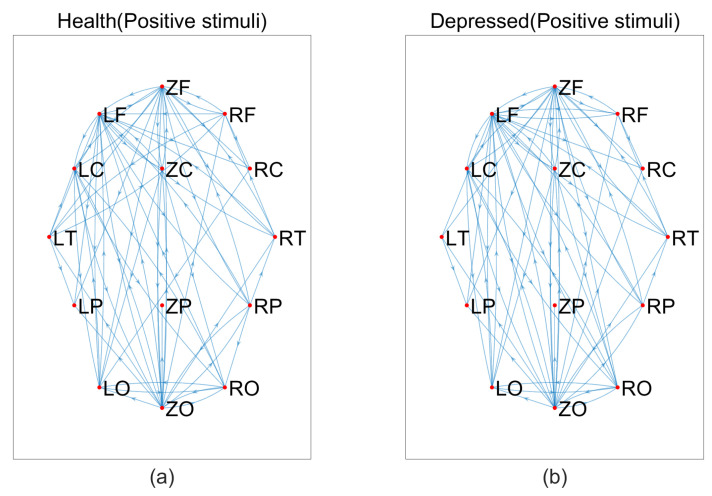
Brain networks in two groups under positive stimuli, (**a**) healthy, (**b**) depressed.

**Figure 7 entropy-25-01330-f007:**
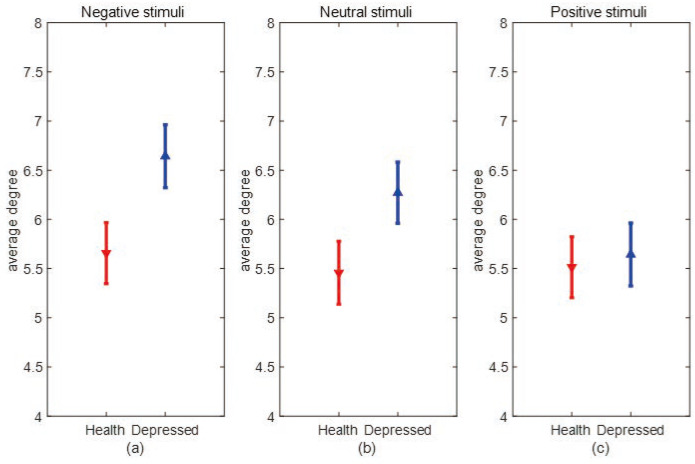
The average degrees of negative, neutral, and positive emotional stimuli in both categories. (**a**) Error bar chart of the average degree between depression patients (blue line) and healthy individuals (red line) under negative stimulation. (**b**) Error bar chart of the average degree between depression patients (blue line) and healthy individuals (red line) under neutral stimulation. (**c**) Error bar chart of the average degree between depression patients (blue line) and healthy individuals (red line) under a positive stimulation.

**Figure 8 entropy-25-01330-f008:**
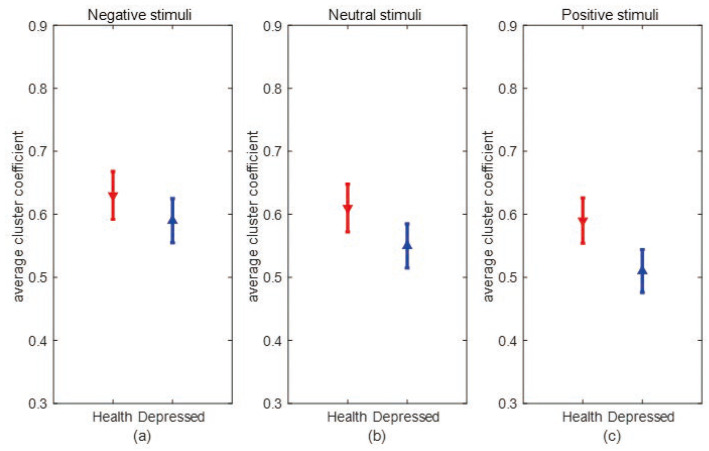
The average clustering coefficients of the two groups under negative, neutral, and positive emotional stimuli. (**a**) Error bar chart of the average clustering coefficient between depression patients (blue line) and healthy individuals (red line) under negative stimulation. (**b**) Error bar chart of the average clustering coefficient between depression patients (blue line) and healthy individuals (red line) under neutral stimulation. (**c**) Error bar chart of the average clustering coefficient between depression patients (blue line) and healthy individuals (red line) under a positive stimulation.

**Figure 9 entropy-25-01330-f009:**
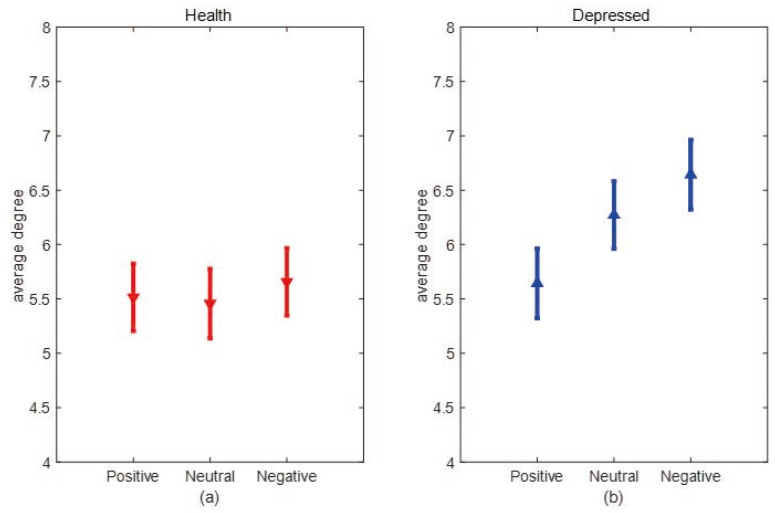
Comparison of the average degree of the same population under different emotional stimuli. (**a**) Error bar chart of the average degree of the healthy group under positive, neutral, and negative stimuli. (**b**) Error bar chart of the average degree of depression patients under positive, neutral, and negative stimuli.

**Figure 10 entropy-25-01330-f010:**
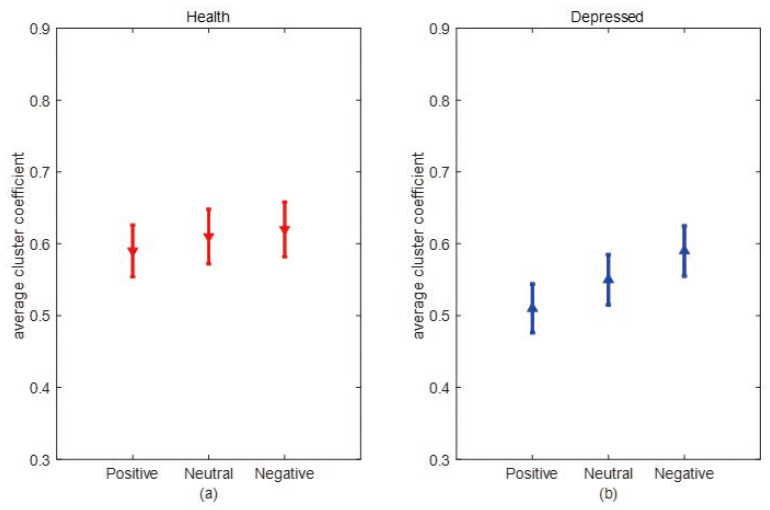
Comparison of average clustering coefficients of the same population under different emotional stimuli. (**a**) Error bar chart of the average clustering coefficient of the healthy group under positive, neutral, and negative stimuli. (**b**) Error bar chart of the average clustering coefficient of depressed patients under positive, neutral, and negative stimuli.

**Figure 11 entropy-25-01330-f011:**
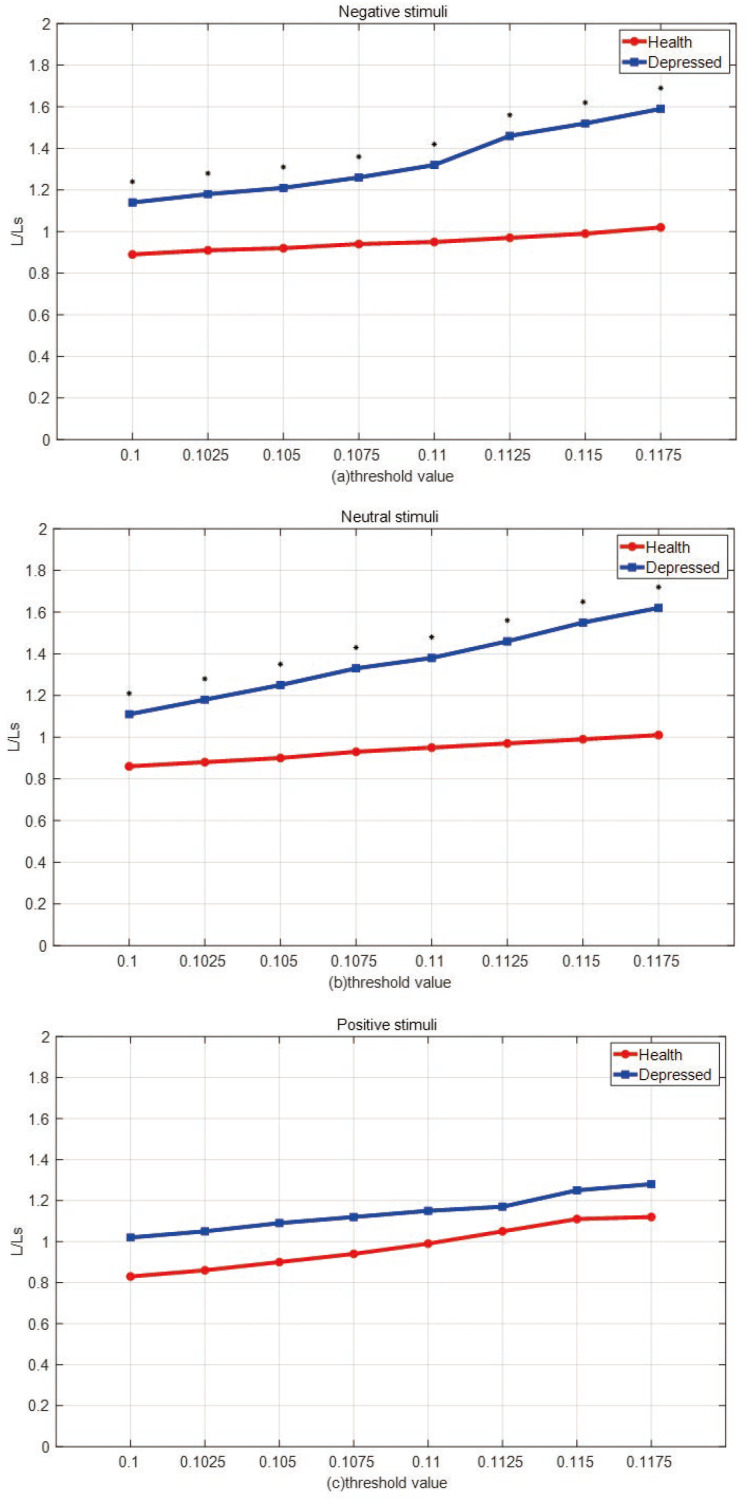
L/Ls values in depressed and healthy individuals under 3 emotional stimuli, * represents *p* < 0.05, (**a**) negative stimuli (**b**) neutral stimuli (**c**) positive stimuli.

**Table 1 entropy-25-01330-t001:** Independent sample *t*-test of the Granger causality mean matrix between 14 brain regions across 2 groups.

	Depression	Normal	t	*p*
Positive stimulus	0.1155 ± 0.0041	0.1158 ± 0.0041	0.040	0.968
Neutral stimulus	0.1198 ± 0.0041	0.1154 ± 0.0041	20.158	0.000
Negative stimulus	0.1211 ± 0.0041	0.1157 ± 0.0041	21.573	0.000

## Data Availability

The data presented in this study are available upon request from the corresponding author.

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
