# Peer review of "Network Analysis of Depression Using Magnetoencephalogram Based on Polynomial Kernel Granger Causality"

_entropy, 2023, doi:10.3390/e25091330_

Round 1

Reviewer 1 Report

This manuscript investigated the changes in functional brain networks in depressed patients using a graph theory-based method. However, the sample size is too small to obtain reliable results, especially for the depressived patient group (n = 5). The writing of the manuscript and presentation of results can be also improved. For example, there are no explanations of each figure panel (a/b/c) in some figure legends.

The writing of the manuscript and presentation of results can be also improved. For example, there are no explanations of each figure panel (a/b/c) in some figure legends.

Reviewer 2 Report

Some of the conclusions of the results presented in this study confirm previous researches on the modifications of small-worldness of brain network in specific pathologies, in this case depression, but also in Alzheimer 's disease and otherforms of dementia. I can suggest to refer to papers on EEG, for example.

Morabito FC, Campolo M, Labate D, Morabito G, Bonanno L, Bramanti A, de Salvo S, Marra A, Bramanti P. A longitudinal EEG study of Alzheimer's disease progression based on a complex network approach. Int J Neural Syst. 2015 Mar;25(2):1550005.

Behrad Soleimani, Proloy Das, I.M. Dushyanthi Karunathilake, Stefanie E. Kuchinsky, Jonathan Z. Simon, Behtash Babadi,

NLGC: Network localized Granger causality with application to MEG directional functional connectivity analysis,NeuroImage,Volume 260,2022, 19496,

Good

Reviewer 3 Report

The paper describes the application of Granger causality in MEG of depressed patients. This is certainly a high end question.

MEG is an expensive research tool and not used to diagnose depression. DX of depression is done clinically (and usually not very difficult). MEG is certainly useful for better understanding of depression as a brain disorder. 

Kernel method for nonlinear Granger causality was developed to detect cause-effect relationships between time series and thus applied to conventional EEG. MEG has a very good temporal solution and is thus prone to an analysis like this. The mathematical foundations and application of network theory is well described.

The authors state correctly that depression is one of the most prevalent psychiatric disorders, but their sample size with n = 5 is rather small. The authors neither describe the type of the depression nor the severity. They do not state, whether the participants of the study were in a depressed mood during the investigation. (If a patient suffers from depression this does not mean that he/she is in depressed a mood all the time.)

Beginning with line 247, they report a plenty of p-values from various t-tests which are obviously not corrected for multiple testing. The various nullhypotheses are not given. It is doubtful whether the reported significant p-values will survive the necessary correction for multiple testing.

They main message of the paper (information exchange between various brain regions differs between depressed patients and healthy controls) is biologically plausible but needs to be confirmed in a larger study.

some  minor spelling and grammar issues

Round 2

Reviewer 1 Report

None

Author Response

Thank you very much for recognizing our revised paper.

Reviewer 3 Report

The authors responded well to my previous suggestions, however, they did not clear my concerns. They acknowledged the fact that they have only five patients, but did not increase the number of patients, and they pointed out that other researchers do not correct their p-values for multiple testing, too. 

moderate editing is recommended

Round 3

Reviewer 3 Report

The authors responded well to my suggestions.

minor editing required